# OpenReview forum: "Evaluating off-the-shell LLMs’  Red-teaming Ability  for Multi-round Jailbreak Attack"
_ICLR.cc/2026/Conference — Submitted to ICLR 2026_

### Official Review · Reviewer_mLcN · 2025-10-23

**Soundness:** 1
**Presentation:** 1
**Contribution:** 1
**Rating:** 0
**Confidence:** 4

**Summary:**

This paper investigates whether off-the-shelf large language models can autonomously conduct multi-round red-teaming tests to uncover safety vulnerabilities in other models. It uses an evaluation framework where attacker models iteratively generate jailbreak prompts against victim models across various harmful content categories. Using automated judging and toxicity scoring, the study assesses the models’ ability to exploit safety weaknesses and analyzes the common techniques they employ. The authors find that existing LLMs possess some inherent red-teaming capability but remain inconsistent, self-restrictive, and inefficient. The paper claims contributions in providing a comprehensive evaluation of such abilities, offering insights for improving automated red-teaming, and suggesting directions for enhancing LLM safety alignment.

**Strengths:**

- Clear experimental framework - A2A setup with CoT and self-reflection is straightforward and well-described
- Multiple evaluation metrics - Goes beyond success rate to include toxic scores, efficiency measures (AFI/AMI), refusal rates, and technique classification

**Weaknesses:**

- The paper’s findings show limited novelty. Its main contribution—evaluating off-the-shelf LLMs as autonomous red-teamers for multi-round jailbreak attacks—largely reiterates established insights from prior work on automated red-teaming and multi-turn adversarial prompting. The results (e.g., that some models can perform jailbreaks while others refuse, or that commercial models use role-play and concealment strategies) are expected and descriptive, not conceptually new. The study introduces no new algorithm, dataset, or theoretical framework; it simply repackages routine comparative evaluations with standard metrics such as attack success rate and toxicity score. Overall, the work’s novelty is incremental and empirical rather than methodological or theoretical
- Overstated claims - calls itself "first comprehensive evaluation" without comprehensive coverage or proper literature review to establish "first"
- Insufficient victim model coverage - only 3 victim models, far too few for "comprehensive evaluation" claims
- Cannot reproduce - missing exact prompts, system messages, sampling parameters, and two victim model identities
- The related work section is severely underdeveloped. It cites only a handful of papers (~6) focused on jailbreak or red-teaming methods, while omitting extensive literature on LLM safety benchmarks, LLM-as-Judge frameworks, and multi-agent red-teaming systems. This narrow coverage fails to contextualize the work within the broader safety and evaluation landscape, leaving the paper’s novelty and positioning unclear.
-  Statistically insufficient - only 10 samples per condition with no significance testing, confidence intervals, or error analysis

**Questions:**

No questions.

---

### Official Review · Reviewer_hiPC · 2025-10-27

**Soundness:** 3
**Presentation:** 1
**Contribution:** 1
**Rating:** 4
**Confidence:** 4

**Summary:**

How well can LLMs red-team other LLMs in a multi-turn scenario? They evaluate 5 categories Security, Criminal, Harassment, Violence, and Sexual and evaluated Qwen-2.5 7B Instruct, DeepSeek v3 and LLama3 70B.

**Strengths:**

Pros
- They ran multi-turn jailbreaks in a simple fashion and evaluated the attacks

**Weaknesses:**

Cons
- This part is confusing to me.  You use the attacker to summarize the attack (which can be an impartial "judge") then have gpt-5 evaluate the success of the summary?
- It's hard to find the exact experimental setup.  A red-team model did 10 rounds against a victim model across the 10 categories.  Was each category done once or multiple times?
- Please just exclude llama 3 70B instead of dashed lines and then explaining that you weren't able to get the model to attack. You can "jailbreak" or harmfully train the model and re-evaluate as a suggestion.
- The limitations seem quite high and aren't properly addressed or noted elsewhere.  Models sometimes highly refuse to do attack a certain category.  Does that mean this category was skipped sometimes? Are we evaluating the same number of attacks across categories?

**Questions:**

Notes:
- Is the title supposed to read "of-the-shelf" instead of "off-the-shell"?
- Line 103 also mentions off-the-shell
- Line 105 missing a citation
- 160 Missing Figure reference
- 187 Doesn't actually say the victim models
- 192 Sentence fragment
- 205 space before comma indicating probably a missing internal reference?
- None of the internal references are working so I'm stopping to point them out

What I'd need to improve my score:
The paper seems rushed and doesn't actually create a nice harmful evaluation platform or instructions.  I was expecting something like HarmBench or similar evaluations setup or infrastructure but this is lacking in quality.

---

### Official Review · Reviewer_hw49 · 2025-11-02

**Soundness:** 3
**Presentation:** 2
**Contribution:** 2
**Rating:** 4
**Confidence:** 4

**Summary:**

This paper evaluates the inherent capability of off-the-shelf Large Language Models (LLMs) to act as red-teamers, specifically for conducting multi-round jailbreak attacks. The authors argue that static, dataset-centric safety evaluations are insufficient as they become outdated and don't transfer well. Using LLMs as red teamers is a scalable alternative, but it's not widely studied whether they can do this "zero-shot" without specialized fine-tuning.

**Strengths:**

- Evaluating multi-turn jailbreaking is useful. Previous works focused on developing agents specifically for this purpose, but knowing the zero-shot ability of various models as red teaming agents is valuable.

**Weaknesses:**

- The attack success standard seems unnecessary. Why does it need to be separate from the activity? Why can't the standard just be "the model carried out the activity"? This is effectively the case for all the examples shown in Figure 1.
- The paper is about evaluating red teaming agents, but it imposes a specific agent scaffold (CoT + self reflection + one jailbreak generated per step). There is a much broader space of red teaming scaffolds that have been developed in this area and that could be explored in this paper. Why not allow arbitrary agent scaffolds?
- The LLM judge was not evaluated by humans for precision and recall. This is a methodological failure that many papers in this area make. I would like to see it corrected in future work, especially papers published at top venues like ICLR.
- There isn't much technical novelty.

**Questions:**

See weaknesses section.

Also, which Llama 3 70B model was used? Llama 3.3 70B instruct or Llama 3.1 70B instruct? This should be specified in the paper.

---

### Meta-Review · Area_Chair_jR2r · 2026-01-01

**Summary:**

This paper investigates whether off-the-shelf LLMs can autonomously conduct multi-turn red-teaming to uncover safety vulnerabilities in other models. However, the AC concurs with the reviewers' concerns, including the limited novelty, an unclear experimental setup, and overstated claims. It is recommended that the authors conduct more rigorous and in-depth experiments and correct the numerous typos throughout the manuscript.

**Reviewer Concerns:**

The authors did not engage in the rebuttal process, and consequently, the reviewers' concerns were not addressed.

**Reviewer Scores:**

The AC thinks the current scores are appropriate.

---

### Decision · Program_Chairs · 2026-01-26

Reject